# Monotonic Differentiable Sorting Networks

**Felix Petersen**[1], **Christian Borgelt**[2], **Hilde Kuehne**[3,4], **Oliver Deussen**[1]
[1]University of Konstanz, [2]University of Salzburg, [3]University of Frankfurt,
[4]IBM-MIT Watson AI Lab,    felix.petersen@uni-konstanz.de

## ABSTRACT

Differentiable sorting algorithms allow training with sorting and ranking supervision, where only the ordering or ranking of samples is known. Various methods have been proposed to address this challenge, ranging from optimal transport-based differentiable Sinkhorn sorting algorithms to making classic sorting networks differentiable. One problem of current differentiable sorting methods is that they are non-monotonic. To address this issue, we propose a novel relaxation of conditional swap operations that guarantees monotonicity in differentiable sorting networks. We introduce a family of sigmoid functions and prove that they produce differentiable sorting networks that are monotonic. Monotonicity ensures that the gradients always have the correct sign, which is an advantage in gradient-based optimization. We demonstrate that monotonic differentiable sorting networks improve upon previous differentiable sorting methods.

## 1 INTRODUCTION

Recently, the idea of end-to-end training of neural networks with ordering supervision via continuous relaxation of the sorting function has been presented by Grover *et al.* [1]. The idea of ordering supervision is that the ground truth order of some samples is known while their absolute values remain unsupervised. This is done by integrating a sorting algorithm in the neural architecture. As the error needs to be propagated in a meaningful way back to the neural network when training with a sorting algorithm in the architecture, it is necessary to use a differentiable sorting function. Several such differentiable sorting functions have been introduced, e.g., by Grover *et al.* [1], Cuturi *et al.* [2], Blondel *et al.* [3], and Petersen *et al.* [4]. In this work, we focus on analyzing differentiable sorting functions [1]–[4] and demonstrate how monotonicity improves differentiable sorting networks [4].

Sorting networks are a family of sorting algorithms that consist of two basic components: so called "wires" (or "lanes") carrying values, and conditional swap operations that connect pairs of wires [5]. An example of such a sorting network is shown in the center of Figure 1. The conditional swap operations swap the values carried by these wires if they are not in the desired order. They allow for fast hardware-implementation, e.g., in ASICs, as well as on highly parallelized general-purpose hardware like GPUs. Differentiable sorting networks [4] continuously relax the conditional swap operations by relaxing their step function to a logistic sigmoid function.

One problem that arises in this context is that using a logistic sigmoid function does not preserve monotonicity of the relaxed sorting operation, which can cause gradients with the wrong sign. In this work, we present a family of sigmoid functions that preserve monotonicity of differentiable sorting networks. These include the cumulative density function (CDF) of the Cauchy distribution, as well as a function that minimizes the error-bound and thus induces the smallest possible approximation error. For all sigmoid functions, we prove and visualize the respective properties and validate their advantages empirically. In fact, by making the sorting function monotonic, it also becomes quasiconvex, which has been shown to produce favorable convergence rates [6]. In Figure 2, we demonstrate monotonicity for different choices of sigmoid functions. As can be seen in Figure 4, existing differentiable sorting operators are either non-monotonic or have an unbounded error.

Following recent work [1], [2], [4], we benchmark our continuous relaxations by predicting values displayed on four-digit MNIST images [7] supervised only by their ground truth order. The evaluation shows that our method outperforms existing relaxations of the sorting function on the four-digit MNIST ordering task as well as the SVHN ranking task.

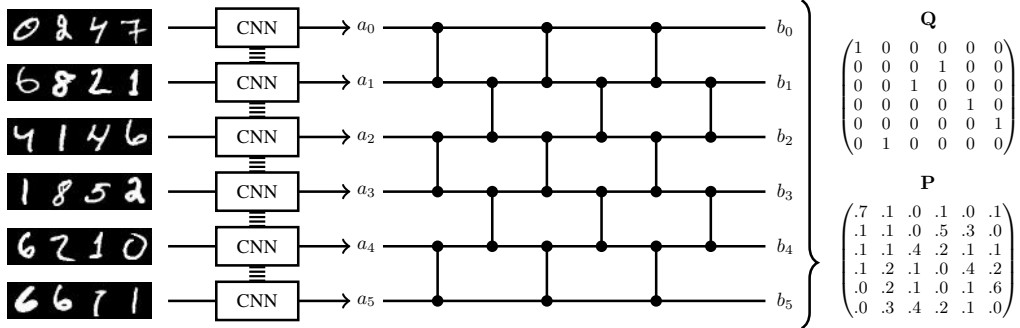

Figure 1: The architecture for training with ordering supervision. Left: input values are fed separately into a Convolutional Neural Network (CNN) that has the same weights for all instances. The CNN maps these values to scalar values $a_0, ..., a_5$. Center: the odd-even sorting network sorts the scalars by parallel conditional swap operations such that all inputs can be propagated to their correct ordered position. Right: It produces a differentiable permutation matrix $\mathbf{P}$. In this experiment, the training objective is the cross-entropy between $\mathbf{P}$ and the ground truth permutation matrix $\mathbf{Q}$. By propagating the error backward through the sorting network, we can train the CNN.

**Contributions.** In this work, we show that sigmoid functions with specific characteristics produce monotonic and error-bounded differentiable sorting networks. We provide theoretical guarantees for these functions and also give the monotonic function that minimizes the approximation error. We empirically demonstrate that the proposed functions improve performance.

## 2 RELATED WORK

Recently, differentiable approximations of the sorting function for weak supervision were introduced by Grover *et al.* [1], Cuturi *et al.* [2], Blondel *et al.* [3], and Petersen *et al.* [4].

In 2019, Grover *et al.* [1] proposed NeuralSort, a continuous relaxation of the argsort operator. A (hard) permutation matrix is a square matrix with entries $0$ and $1$ such that every row and every column sums up to $1$, which defines the permutation necessary to sort a sequence. Grover *et al.* relax hard permutation matrices by approximating them as unimodal row-stochastic matrices. This relaxation allows for gradient-based stochastic optimization. On various tasks, including sorting four-digit MNIST numbers, they benchmark their relaxation against the Sinkhorn and Gumbel-Sinkhorn approaches proposed by Mena *et al.* [8].

Cuturi *et al.* [2] follow this idea and approach differentiable sorting by smoothed ranking and sorting operators using optimal transport. As the optimal transport problem alone is costly, they regularize it and solve it using the Sinkhorn algorithm [9]. By relaxing the permutation matrix, which sorts a sequence of scalars, they also train a scalar predictor of values displayed by four-digit numbers while supervising their relative order only.

Blondel *et al.* [3] cast the problem of sorting and ranking as a linear program over a permutahedron. To smooth the resulting discontinuous function and provide useful derivatives, they introduce a strongly convex regularization. They evaluate the proposed approach in the context of top-k classification and label ranking accuracy via a soft Spearman's rank correlation coefficient.

Recently, Petersen *et al.* [4] proposed differentiable sorting networks, a differentiable sorting operator based on sorting networks with differentiably relaxed conditional swap operations. Differentiable sorting networks achieved a new state-of-the-art on both the four-digit MNIST sorting benchmark and the SVHN sorting benchmark. Petersen *et al.* [10] also proposed a general method for continuously relaxing algorithms via logistic distributions. They apply it, i.a., to the bubble sort algorithm and benchmark in on the MNIST sorting benchmark.

**Applications and Broader Impact.** In the domain of recommender systems, Lee *et al.* [11] propose differentiable ranking metrics, and Swezey *et al.* [12] propose PiRank, a learning-to-rank method using differentiable sorting. Other works explore differentiable sorting-based top-k for applications such as differentiable image patch selection [13], differentiable k-nearest-neighbor [1], [14], top-k attention for machine translation [14], and differentiable beam search methods [14], [15].

## 3 BACKGROUND: SORTING NETWORKS

Sorting networks have a long tradition in computer science since the 1950s [5]. They are highly parallel data-oblivious sorting algorithms. They are based on so-called conditional pairwise swap operators that map two inputs to two outputs and ensure that these outputs are in a specific order. This is achieved by simply passing through the inputs if they are already in the desired order and swapping them otherwise. The order of executing conditional swaps is independent of the input values, which makes them data-oblivious. Conditional swap operators can be implemented using only $\min$ and $\max$. That is, if the inputs are $a$ and $b$ and the outputs $a'$ and $b'$, a swap operator ensures $a' \leq b'$ and can easily be formalized as $a' = \min(a, b)$ and $b' = \max(a, b)$. Examples of sorting nets are the odd-even network [16], which alternatingly swaps odd and even wires with their successors, and the bitonic network [17], which repeatedly merges and sorts bitonic sequences. While the odd-even network requires $n$ layers, the bitonic network uses the divide-and-conquer principle to sort within only $(\log_2 n)(1 + \log_2 n)/2$ layers.

Note that, while they are similar in name, sorting networks are *not* neural networks that sort.

### 3.1 DIFFERENTIABLE SORTING NETWORKS

In the following, we recapitulate the core concepts of differentiable sorting networks [4]. An example of an odd-even sorting network is shown in the center of Figure 1. Here, odd and even neighbors are conditionally swapped until the entire sequence is sorted. Each conditional swap operation can be defined in terms of $\min$ and $\max$ as detailed above. These operators can be relaxed to differentiable $min$ and $max$. Note that we denote the differentiable relaxations in *italic* font and their hard counterparts in roman font. Note that the differentiable relaxations $min$ and $max$ are different from the commonly used $\mathrm{softmin}$ and $\mathrm{softmax}$, which are relaxations of $\mathrm{argmin}$ and $\mathrm{argmax}$ [18].

One example for such a relaxation of $\min$ and $\max$ is the logistic relaxation

$$min_\sigma(a, b) = a \cdot \sigma(b - a) + b \cdot \sigma(a - b) \quad \text{and} \quad max_\sigma(a, b) = a \cdot \sigma(a - b) + b \cdot \sigma(b - a) \quad (1)$$

where $\sigma$ is the logistic sigmoid function with inverse temperature $\beta > 0$:

$$\sigma : x \mapsto \frac{1}{1 + e^{-\beta x}}. \tag{2}$$

Any layer of a sorting network can also be represented as a relaxed and doubly-stochastic permutation matrix. Multiplying these (layer-wise) permutation matrices yields a (relaxed) total permutation matrix $\mathbf{P}$. Multiplying $\mathbf{P}$ with an input $x$ yields the differentiably sorted vector $\hat{x} = \mathbf{P}x$, which is also the output of the differentiable sorting network. Whether it is necessary to compute $\mathbf{P}$, or whether $\hat{x}$ suffices, depends on the specific application. For example, for a cross-entropy ranking / sorting loss as used in the experiments in Section 6, $\mathbf{P}$ can be used to compute the cross-entropy to a ground truth permutation matrix $\mathbf{Q}$.

In the next section, we build on these concepts to introduce monotonic differentiable sorting networks, i.e., all differentiably sorted outputs $\hat{x}$ are non-decreasingly monotonic in all inputs $x$.

## 4 MONOTONIC DIFFERENTIABLE SORTING NETWORKS

In this section, we start by introducing definitions and building theorems upon them. In Section 4.2, we use these definitions and properties to discuss different relaxations of sorting networks.

### 4.1 THEORY

We start by defining sigmoid functions and will then use them to define continuous conditional swaps.

**Definition 1** (Sigmoid Function)**.** *We define a (unipolar) sigmoid (i.e., s-shaped) function as a continuous monotonically non-decreasing odd-symmetric (around $\frac{1}{2}$) function $f$ with*

$$f : \mathbb{R} \to [0, 1] \quad \text{with} \quad \lim_{x \to -\infty} f(x) = 0 \quad \text{and} \quad \lim_{x \to \infty} f(x) = 1.$$

**Definition 2** (Continuous Conditional Swaps). *Following [4], we define a continuous conditional swap in terms of a sigmoid function $f : \mathbb{R} \to [0, 1]$ as*

$$min_f(a, b) = a \cdot f(b - a) + b \cdot f(a - b), \qquad max_f(a, b) = a \cdot f(a - b) + b \cdot f(b - a), \quad (3)$$

$$argmin_f(a, b) = (\ f(b - a), \quad f(a - b)\ ), \quad argmax_f(a, b) = (\ f(a - b), \quad f(b - a)\ ). \quad (4)$$

We require a continuous odd-symmetric sigmoid function to preserve most of the properties of $min$ and $max$ while also making $argmin$ and $argmax$ continuous as shown in Supplementary Material B. In the following, we establish doubly-stochasticity and differentiability of $\mathbf{P}$, which are important properties for differentiable sorting and ranking operators.

**Lemma 3** (Doubly-Stochasticity and Differentiability of $\mathbf{P}$). *(i) The relaxed permutation matrix $\mathbf{P}$, produced by a differentiable sorting network, is doubly-stochastic. (ii) $\mathbf{P}$ has the same differentiability as $f$, e.g., if $f$ is continuously differentiable in the input, $\mathbf{P}$ will be continuously differentiable in the input to the sorting network. If $f$ is differentiable almost everywhere (a.e.), $\mathbf{P}$ will be diff. a.e.*

*Proof.* (i) For each conditional swap between two elements $i, j$, the relaxed permutation matrix is $1$ at the diagonal except for rows $i$ and $j$: at points $i, i$ and $j, j$ the value is $v \in [0, 1]$, at points $i, j$ and $j, i$ the value is $1 - v$ and all other entries are $0$. This is doubly-stochastic as all rows and columns add up to $1$ by construction. As the product of doubly-stochastic matrices is doubly-stochastic, the relaxed permutation matrix $\mathbf{P}$, produced by a differentiable sorting network, is doubly-stochastic.

(ii) The composition of differentiable functions is differentiable and the addition and multiplication of differentiable functions is also differentiable. Thus, a sorting network is differentiable if the employed sigmoid function is differentiable. "Differentiable" may be replaced with any other form of differentiability, such as "differentiable a.e." □

Now that we have established the ingredients to differentiable sorting networks, we can focus on the monotonicity of differentiable sorting networks.

**Definition 4** (Monotonic Continuous Conditional Swaps). *We say $f$ produces monotonic conditional swaps if $min_f(x, 0)$ is non-decreasingly monotonic in $x$, i.e., $min'_f(x, 0) \geq 0$ for all $x$.*

It is sufficient to define it w.l.o.g. in terms of $min_f(x, 0)$ due to its commutativity, stability, and odd-symmetry of the operators (cf. Supplementary Material B).

**Theorem 5** (Monotonicity of Continuous Conditional Swaps). *A continuous conditional swap (in terms of a differentiable sigmoid function $f$) being non-decreasingly monotonic in all arguments and outputs requires that the derivative of $f$ decays no faster than $1/x^2$, i.e.,*

$$f'(x) \in \Omega\left(\frac{1}{x^2}\right). \quad (5)$$

*Proof.* We show that Equation 5 is a necessary criterion for monotonicity of the conditional swap. Because $f$ is a continuous sigmoid function with $f : \mathbb{R} \to [0, 1]$, $min_f(x, 0) = f(-x) \cdot x > 0$ for some $x > 0$. Thus, montononicity of $min_f(x, 0)$ implies $\limsup_{x \to \infty} min_f(x, 0) > 0$ (otherwise the value would decrease again from a value $> 0$.) Thus,

$$\lim_{x \to \infty} min_f(x, 0) = \lim_{x \to \infty} f(-x) \cdot x = \lim_{x \to \infty} \frac{f(-x)}{1/x} \overset{\text{(L'Hôpital's rule)}}{=} \lim_{x \to \infty} \frac{-f'(-x)}{-1/x^2} \quad (6)$$

$$= \lim_{x \to \infty} \frac{f'(-x)}{1/x^2} = \lim_{x \to \infty} \frac{f'(x)}{1/x^2} = \limsup_{x \to \infty} \frac{f'(x)}{1/x^2} > 0 \iff f'(x) \in \Omega\left(\frac{1}{x^2}\right). \quad (7)$$

assuming $\lim_{x \to \infty} \frac{f'(x)}{1/x^2}$ exists. Otherwise, it can be proven analogously via a proof by contradiction. □

**Corollary 6** (Monotonic Sorting Networks). *If the individual conditional swaps of a sorting network are monotonic, the sorting network is also monotonic.*

*Proof.* If single layers $g, h$ are non-decreasingly monotonic in all arguments and outputs, their composition $h \circ g$ is also non-decreasingly monotonic in all arguments and outputs. Thus, a network of arbitrarily many layers is non-decreasingly monotonic. □

Above, we formalized the property of monotonicity. Another important aspect is whether the error of the differentiable sorting network is bounded. It is very desirable to have a bounded error because without bounded errors the result of the differentiable sorting network diverges from the result of the hard sorting function. Minimizing this error is desirable.

**Definition 7** (Error-Bounded Continuous Conditional Swaps). *A continuous conditional swap has a bounded error if and only if $\sup_x min_f(x, 0) = c$ is finite. The continuous conditional swap is therefore said to have an error bounded by $c$.*

It is sufficient to define it w.l.o.g. in terms of $min_f(x, 0)$ due to its commutativity, stability, and odd-symmetry of the operators (cf. Supplementary Material B). In general, for better comparability between functions, we assume a Lipschitz continuous function $f$ with Lipschitz constant $1$.

**Theorem 8** (Error-Bounds of Continuous Conditional Swaps). *(i) A differentiable continuous conditional swap has a bounded error if*

$$f'(x) \in \mathcal{O}\left(\frac{1}{x^2}\right). \tag{8}$$

*(ii) If it is additionally monotonic, the error-bound can be found as $\lim_{x \to \infty} min_f(x, 0)$ and additionally the error is bound only if Equation 8 holds.*

*Proof.* (i) W.l.o.g. we consider $x > 0$. Let $g(z) := f(-1/z), g(0) = 0$. Thus, $g'(z) = 1/z^2 \cdot f'(-1/z) \leq c$ according to Equation 8. Thus, $g(z) = g(0) + \int_0^z g'(t)dt \leq c \cdot z$. Therefore, $f(-1/z) \leq c \cdot z \implies 1/z \cdot f(-1/z) \leq c$ and with $x = 1/z \implies x \cdot f(-x) = min_f(x, 0) \leq c$.

(ii) Let $min_f(x, 0)$ be monotonic and bound by $min_f(x, 0) \leq c$. For $x > 0$ and $h(x) := min_f(x, 0)$,

$$h'(x) = -x \cdot f'(-x) + f(-x) \implies x^2 f'(-x) = \underbrace{-xh'(x)}_{\leq 0} + x \cdot f(-x) \leq x \cdot f(-x) \leq c. \tag{9}$$

And thus $f'(x) \in \mathcal{O}\left(\frac{1}{x^2}\right)$. $\qquad\square$

**Theorem 9** (Error-Bounds of Diff. Sorting Networks). *If the error of individual conditional swaps of a sorting network is bounded by $\epsilon$ and the network has $\ell$ layers, the total error is bounded by $\epsilon \cdot \ell$.*

*Proof.* For the proof, cf. Supplementary Material D. $\qquad\square$

**Discussion.** Monotonicity is highly desirable as otherwise adverse effects such as an input requiring to be decreased to increase the output can occur. In gradient-based training, non-mononicity is problematic as it produces gradients with the opposite sign. In addition, as monotonicity is also given in hard sorting networks, it is desirable to preserve this property in the relaxation. Further, monotonic differentiable sorting networks are quasiconvex and quasiconcave as any monotonic function is both quasiconvex and quasiconcave, which leads to favorable convergence rates [6]. Bounding and reducing the deviation from its hard counterpart reduces the relaxation error, and thus is desirable.

## 4.2 SIGMOID FUNCTIONS

Above, we have specified the space of functions for the differentiable swap operation, as well as their desirable properties. In the following, we discuss four notable candidates as well as their properties. The properties of these functions are visualized in Figures 2 and 3 and an overview over their properties is given in Table 1.

**Logistic distributions.** The first candidate is the logistic sigmoid function (the CDF of a logistic distribution) as proposed in [4]:

$$\sigma(x) = \text{CDF}_{\mathcal{L}}\left(\beta x\right) = \frac{1}{1 + e^{-\beta x}} \tag{10}$$

This function is the de-facto default sigmoid function in machine learning. It provides a continuous, error-bounded, and Lipschitz continuous conditional swap. However, for the logistic function, monotonicity is not given, as displayed in Figure 2.

Table 1: For each function, we display the *function*, its *derivative*, and indicate whether the respective relaxed sorting network is *monotonic* and has a *bounded error*.

| Function | $f$ (CDF) | $f'$ (PDF) | Eq. | Mono. | Bounded Error |
|---|---|---|---|---|---|
| $\sigma$ | | | (10) | ✗ | ✓ $(\approx .0696/\alpha)$ |
| $f_{\mathcal{R}}$ | | | (11) | ✓ | ✓ $(1/4/\alpha)$ |
| $f_{\mathcal{C}}$ | | | (12) | ✓ | ✓ $(1/\pi^2/\alpha)$ |
| $f_{\mathcal{O}}$ | | | (13) | ✓ | ✓ $(1/16/\alpha)$ |

**Reciprocal Sigmoid Function.** To obtain a function that yields a monotonic as well as error-bound differentiable sorting network, a necessary criterion is $f'(x) \in \Theta(1/x^2)$ (the intersection of Equations 5 and 8.) A natural choice is, therefore, $f'_{\mathcal{R}}(x) = \frac{1}{(2|x|+1)^2}$, which produces

$$f_{\mathcal{R}}(x) = \int_{-\infty}^{x} \frac{1}{(2\beta|t|+1)^2} dt = \frac{1}{2} \frac{2\beta x}{1+2\beta|x|} + \frac{1}{2}. \tag{11}$$

$f_{\mathcal{R}}$ fulfills all criteria, i.e., it is an adequate sigmoid function and produces monotonic and error-bound conditional swaps. It has an $\epsilon$-bounded-error of $\epsilon = 0.25$. It is also an affine transformation of the elementary bipolar sigmoid function $x \mapsto \frac{x}{|x|+1}$. Properties of this function are visualized in Table 1 and Figures 2 and 3. Proofs for monotonicity can be found in Supplementary Material D.

**Cauchy distributions.** By using the CDF of the Cauchy distribution, we maintain montonicity while reducing the error-bound to $\epsilon = 1/\pi^2 \approx 0.101$. It is defined as

$$f_{\mathcal{C}}(x) = \text{CDF}_{\mathcal{C}}\left(\beta x\right) = \frac{1}{\pi} \int_{-\infty}^{x} \frac{\beta}{1+(\beta t)^2} dt = \frac{1}{\pi} \arctan\left(\beta x\right) + \frac{1}{2} \tag{12}$$

In the experimental evaluation, we find that tightening the error improves the performance.

**Optimal Monotonic Sigmoid Function.** At this point, we are interested in the monotonic swap operation that minimizes the error-bound. Here, we set 1-*Lipschitz continuity* again as a requirement to make different relaxations of conditional swaps comparable. We show that $f_{\mathcal{O}}$ is the best possible sigmoid function achieving an error-bound of only $\epsilon = 1/16$

**Theorem 10** (Optimal Sigmoid Function). *The optimal sigmoid function minimizing the error-bound, while producing a monotonic and* 1-*Lipschitz continuous (with $\beta = 1$) conditional swap operation, is*

$$f_{\mathcal{O}}(x) = \begin{cases} -\frac{1}{16\beta x} & \text{if } \beta x < -\frac{1}{4}, \\ 1 - \frac{1}{16\beta x} & \text{if } \beta x > +\frac{1}{4}, \\ \beta x + \frac{1}{2} & \text{otherwise.} \end{cases} \tag{13}$$

*Proof.* Given the above conditions, the optimal sigmoid function is uniquely determined and can easily be derived as follows: Due to stability, it suffices to consider $min_f(x, 0) = x \cdot f(-x)$ or $max_f(0, x) = -x \cdot f(x)$. Due to symmetry and inversion, it suffices to consider $min_f(x, 0) = x \cdot f(-x)$ for $x > 0$.

Since $\min(x, 0) = 0$ for $x > 0$, we have to choose $f$ in such a way as to make $min_f(x, 0) = x \cdot f(-x)$ as small as possible, but not negative. For this, $f(-x)$ must be made as small as possible. Since we know that $f(0) = \frac{1}{2}$ and we are limited to functions $f$ that are Lipschitz continuous with $\alpha = 1$, $f(-x)$ cannot be made smaller than $\frac{1}{2} - x$, and hence $min_f(x, 0)$ cannot be made smaller than $x \cdot \left(\frac{1}{2} - x\right)$. To make $min_f(x, 0)$ as small as possible, we have to follow $x \cdot \left(\frac{1}{2} - x\right)$ as far as possible (i.e., to values $x$ as large as possible). Monotonicity requires that this function can be followed only up to $x = \frac{1}{4}$, at which point we have $min_f(\frac{1}{4}, 0) = \frac{1}{4}\left(\frac{1}{2} - \frac{1}{4}\right) = \frac{1}{16}$. For larger $x$, that is, for $x > \frac{1}{4}$, the value of $x \cdot \left(\frac{1}{2} - x\right)$ decreases again and hence the functional form of the sigmoid function $f$ has to change at $x = \frac{1}{4}$ to remain monotonic.

The best that can be achieved for $x > \frac{1}{4}$ is to make it constant, as it must not decrease (due to monotonicity) and should not increase (to minimize the deviation from the crisp / hard version). That is, $min_f(x, 0) = \frac{1}{16}$ for $x > \frac{1}{4}$. It follows $x \cdot f(-x) = \frac{1}{16}$ and hence $f(-x) = \frac{1}{16x}$ for $x > \frac{1}{4}$. Note that, if the transition from the linear part to the hyperbolic part were at $|x| < \frac{1}{4}$, the function would not be Lipschitz continuous with $\alpha = 1$. $\qquad\square$

An overview of the selection of sigmoid functions we consider is shown in Table 1. Note how $f_\mathcal{R}$, $f_\mathcal{C}$ and $f_\mathcal{O}$ in this order get closer to $x + \frac{1}{2}$ (the gray diagonal line) and hence steeper in their middle part. This is reflected by a widening region of values of the derivatives that are close to or even equal to 1.

Table 1 also indicates whether a sigmoid function yields a *monotonic* swap operation or not, which is visualized in Figure 2: clearly $\sigma$-based sorting networks are not monotonic, while all others are. It also states whether the *error is bounded*, which for a monotonic swap operation means $\lim_{x \to \infty} min_f(x, 0) < \infty$, and gives their bound relative to the Lipschitz constant $\alpha$.

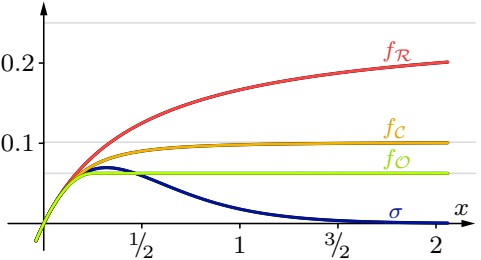

Figure 2: $min_f(x, 0)$ for different sigmoid functions $f$; color coding as in Table 1.

Figure 3 displays the loss for a sorting network with $n = 3$ inputs. We project the hexagon-shaped 3-value permutahedron onto the $x$-$y$-plane, while the $z$-axis indicates the loss. Note that, at the rightmost point $(1, 2, 3)$, the loss is 0 because all elements are in the correct order, while at the left front $(2, 3, 1)$ and rear $(3, 1, 2)$ the loss is at its maximum because all elements are at the wrong positions. Along the red center line, the loss rises logarithmic for the optimal sigmoid function on the right. Note that the monotonic sigmoid functions produce a loss that is larger when more elements are in the wrong order. For the logistic function, $(3, 2, 1)$ has the same loss as $(2, 3, 1)$ even though one of the ranks is correct at $(3, 2, 1)$, while for $(2, 3, 1)$ all three ranks are incorrect.

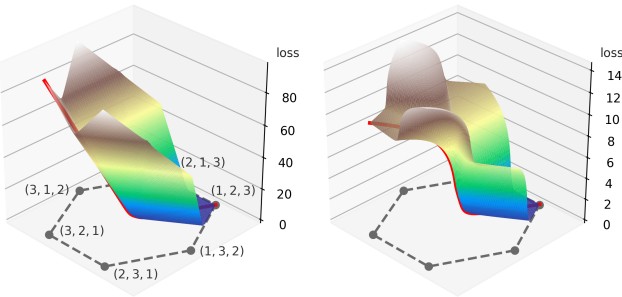

Figure 3: Loss for a 3-wire odd-even sorting network, drawn over a permutahedron projected onto the $x$-$y$-plane. For logistic sigmoid (left) and optimal sigmoid (right).

## 5    MONOTONICITY OF OTHER DIFFERENTIABLE SORTING OPERATORS

For the special case of $n = 2$, i.e., for sorting two elements, *NeuralSort* [1] and *Relaxed Bubble sort* [10] are equivalent to differentiable sorting networks with the logistic sigmoid function. Thus, it is non-monotonic, as displayed in Figure 4.

For the *Sinkhorn sorting algorithm* [2], we can simply construct an example of non-monotonicity by keeping one value fixed, e.g., at zero, and varying the second value ($x$) as in Figure 4 and displaying the minimum. Notably, for the case of $n = 2$, this function is numerically equal to NeuralSort and differentiable sorting networks with the logistic function.

For *fast sort* [3], we follow the same principle and find that it is indeed monotonic (in this example); however, the error is unbounded, which is undesirable.

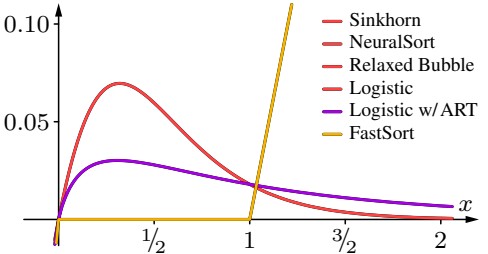

Figure 4: $min(x, 0)$ for Sinkhorn sort (red), NeuralSort (red), Relaxed Bubble sort (red), diffsort with logistic sigmoid (red), diffsort with activation replacement trick (purple), and Fast Sort (orange).

For *differentiable sorting networks*, Petersen *et al.* [4] proposed to extend the sigmoid function by the

activation replacement trick, which avoids extensive blurring as well as vanishing gradients. They apply the activation replacement trick $\varphi$ before feeding the values into the logistic sigmoid function; thus, the sigmoid function is effectively $\sigma \circ \varphi$. Here, $\varphi : x \mapsto \frac{x}{|x|^\lambda + \epsilon}$ where $\lambda \in [0,1]$ and $\epsilon \approx 10^{-10}$. Here, the asymptotic character of $\sigma \circ \varphi$ does not fulfill the requirement set by Theorem 5, and is thereby non-monotonic as also displayed in Figure 4 (purple).

We summarize monotonicity and error-boundness for all differentiable sorting functions in Table 2.

Table 2: For each differentiable sorting operator, whether it is monotonic (M), and whether it has a bounded error (BE).

| Method | | M | BE |
|---|---|---|---|
| NeuralSort | | ✗ | – |
| Sinkhorn Sort | | ✗ | – |
| Fast Sort | | ✓ | ✗ |
| Relaxed Bubble Sort | | ✗ | – |
| Diff. Sorting Networks | $\sigma$ | ✗ | ✓ |
| Diff. Sorting Networks | $\sigma \circ \varphi$ | ✗ | ✓ |
| Diff. Sorting Networks | $f_{\mathcal{R}}$ | ✓ | ✓ |
| Diff. Sorting Networks | $f_{\mathcal{C}}$ | ✓ | ✓ |
| Diff. Sorting Networks | $f_{\mathcal{O}}$ | ✓ | ✓ |

## 6 EMPIRICAL EVALUATION

To evaluate the properties of the proposed function as well as their practical impact in the context of sorting supervision, we evaluate them with respect to two standard benchmark datasets. The MNIST sorting dataset [1]–[4] consists of images of numbers from 0000 to 9999 composed of four MNIST digits [7]. Here, the task is training a network to produce a scalar output value for each image such that the ordering of the outputs follows the respective ordering of the images. Specifically, the metrics here are the proportion of full rankings correctly identified, and the proportion of individual element ranks correctly identified [1]. The same task can also be extended to the more realistic SVHN [19] dataset with the difference that the images are already multi-digit numbers as shown in [4].

**Comparison to the State-of-the-Art.** We first compare the proposed functions to other state-of-the-art approaches using the same network architecture and training setup as used in previous works, as well as among themselves. The respective hyperparameters for each setting can be found in Supplementary Material A. We report the results in Table 3. The proposed monotonic differentiable sorting networks outperform current state-of-the-art methods by a considerable margin. Especially for those cases where more samples needed to be sorted, the gap between monotonic sorting nets and other techniques grows with larger $n$. The computational complexity of the proposed method depends on the employed sorting network architecture leading to a complexity of $\mathcal{O}(n^3)$ for odd-even networks and a complexity of $\mathcal{O}(n^2 \log^2 n)$ for bitonic networks because all of the employed sigmoid functions can be computed in closed form. This leads to the same runtime as in [4].

Comparing the three proposed functions among themselves, we observe that for odd-even networks on MNIST, the error-optimal function $f_{\mathcal{O}}$ performs best. This is because here the approximation error is small. However, for the more complex bitonic sorting networks, $f_{\mathcal{C}}$ (Cauchy) performs better than $f_{\mathcal{O}}$. This is because $f_{\mathcal{O}}$ does not provide a higher-order smoothness and is only $C^1$ smooth, while the Cauchy function $f_{\mathcal{C}}$ is analytic and $C^\infty$ smooth.

Table 3: Results on the four-digit MNIST and SVHN tasks using the same architecture as previous works [1]–[4]. The metric is the proportion of rankings correctly identified, and the value in parentheses is the proportion of individual element ranks correctly identified. All results are averaged over 5 runs. SVHN w/ $n = 32$ is omitted to reduce the carbon impact of the evaluation.

| MNIST | $n = 3$ | $n = 5$ | $n = 7$ | $n = 9$ | $n = 15$ | $n = 32$ | $n = 16$ (bitonic) | $n = 32$ (bitonic) |
|---|---|---|---|---|---|---|---|---|
| NeuralSort | 91.9 (94.5) | 77.7 (90.1) | 61.0 (86.2) | 43.4 (82.4) | 9.7 (71.6) | 0.0 (38.8) | — | — |
| Sinkhorn Sort | 92.8 (95.0) | 81.1 (91.7) | 65.6 (88.2) | 49.7 (84.7) | 12.6 (74.2) | 0.0 (41.2) | — | — |
| Fast Sort & Rank | 90.6 (93.5) | 71.5 (87.2) | 49.7 (81.3) | 29.0 (75.2) | 2.8 (60.9) | — | — | — |
| Diffsort (Logistic) | 92.0 (94.5) | 77.2 (89.8) | 54.8 (83.6) | 37.2 (79.4) | 4.7 (62.3) | 0.0 (56.3) | 10.8 (72.6) | 0.3 (63.2) |
| Diffsort (Log. w/ ART) | 94.3 (96.1) | 83.4 (92.6) | 71.6 (90.0) | 56.3 (86.7) | 23.5 (79.4) | 0.5 (64.9) | 19.0 (77.5) | 0.8 (63.0) |
| $f_{\mathcal{R}}$ : Reciprocal Sigmoid | 94.4 (96.1) | 85.0 (93.3) | 73.4 (90.7) | 60.8 (88.1) | 30.2 (81.9) | 1.0 (66.8) | 28.7 (82.1) | 1.3 (68.0) |
| $f_{\mathcal{C}}$ : Cauchy CDF | 94.2 (96.0) | 84.9 (93.2) | 73.3 (90.5) | 63.8 (89.1) | 31.1 (82.2) | 0.8 (63.3) | 29.0 (82.1) | 1.6 (68.1) |
| $f_{\mathcal{O}}$ : Optimal Sigmoid | 94.6 (96.3) | 85.0 (93.3) | 73.6 (90.7) | 62.2 (88.5) | 31.8 (82.3) | 1.4 (67.9) | 28.4 (81.9) | 1.4 (67.7) |
| SVHN | $n = 3$ | $n = 5$ | $n = 7$ | $n = 9$ | $n = 15$ | — | $n = 16$ (bitonic) | — |
| Diffsort (Logistic) | 76.3 (83.2) | 46.0 (72.7) | 21.8 (63.9) | 13.5 (61.7) | 0.3 (45.9) | — | 1.2 (50.6) | — |
| Diffsort (Log. w/ ART) | 83.2 (88.1) | 64.1 (82.1) | 43.8 (76.5) | 24.2 (69.6) | 2.4 (56.8) | — | 3.4 (59.2) | — |
| $f_{\mathcal{R}}$ : Reciprocal Sigmoid | 85.7 (89.8) | 68.8 (84.2) | 53.3 (80.0) | 40.0 (76.3) | 13.2 (66.0) | — | 11.5 (64.9) | — |
| $f_{\mathcal{C}}$ : Cauchy CDF | 85.5 (89.6) | 68.5 (84.1) | 52.9 (79.8) | 39.9 (75.8) | 13.7 (66.0) | — | 12.2 (65.6) | — |
| $f_{\mathcal{O}}$ : Optimal Sigmoid | 86.0 (90.0) | 67.5 (83.5) | 53.1 (80.0) | 39.1 (76.0) | 13.2 (66.3) | — | 10.6 (66.8) | — |

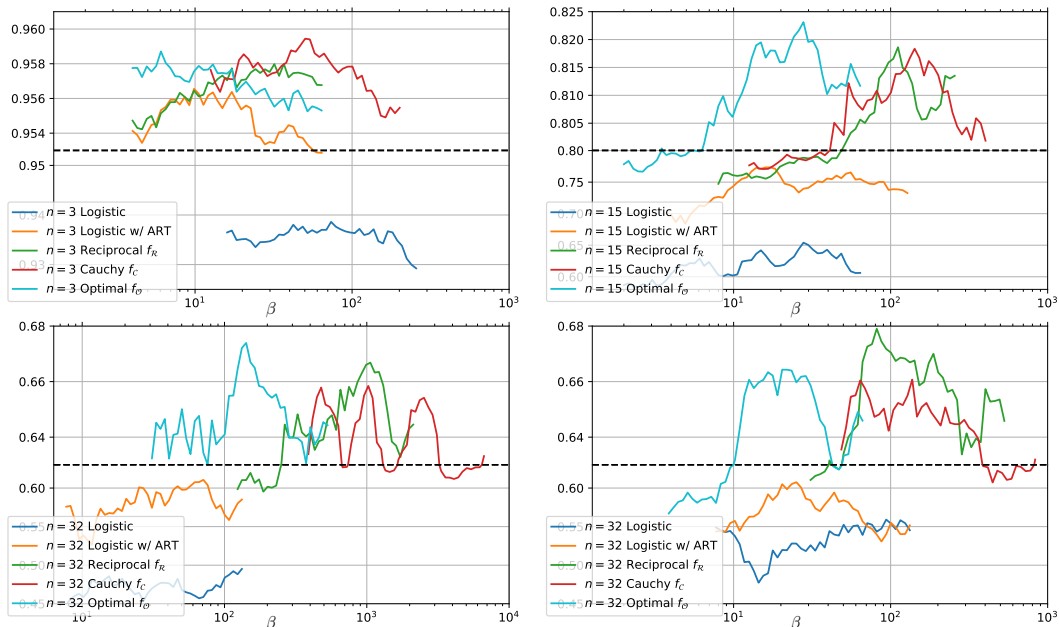

Figure 5: Evaluating different sigmoid functions on the sorting MNIST task for ranges of different inverse temperatures $\beta$. The metric is the proportion of individual element ranks correctly identified. In all settings, the monotonic sorting networks clearly outperform the non-monotonic ones. Top: Odd-Even sorting networks with $n = 3$ (left) and $n = 15$ (right). Bottom: $n = 32$ with an Odd-Even (left) and a Bitonic network (right). For small $n$, such as 3, Cauchy performs best because it has a low error but is smooth at the same time. For larger $n$, such as 15 and 32, the optimal sigmoid function (wrt. error) $f_{\mathcal{O}}$ performs better because it, while not being smooth, has the smallest possible approximation error which is more important for deeper networks. For the bitonic network with its more complex structure at $n = 32$ (bottom right), the reciprocal sigmoid $f_{\mathcal{R}}$ performs best.

**Evaluation of Inverse Temperature $\beta$.**  To further understand the behavior of the proposed monotonic functions compared to the logistic sigmoid function, we evaluate all sigmoid functions for different inverse temperatures $\beta$ during training. We investigate four settings: odd-even networks for $n \in \{3, 15, 32\}$ and a bitonic sorting network with $n = 32$ on the MNIST data set. Notably, there are 15 layers in the bitonic sorting networks with $n = 32$, while the odd-even networks for $n = 15$ also has 15 layers. We display the results of this evaluation in Figure 5. In Supplementary Material C, we show an analogous figure with additional settings. Note that here, we train for only $50\%$ of the training steps compared to Table 3 to reduce the computational cost.

We observe that the optimal inverse temperature depends on the number of layers, rather than the overall number of samples $n$. This can be seen when comparing the peak accuracy of each function for the odd-even sorting network for different $n$ and thus for different numbers of layers. The bitonic network for $n = 32$ (bottom right) has the same number of layers as $n = 15$ in the odd-even network (top right). Here, the peak performances for each sigmoid function fall within the same range, whereas the peak performances for the odd-even network for $n = 32$ (bottom left) are shifted almost an order of magnitude to the right. For all configurations, the proposed sigmoid functions for monotonic sorting networks improve over the standard logistic sigmoid function, as well as the ART.

The source code of this work is publicly available at github.com/Felix-Petersen/diffsort.

# 7 CONCLUSION

In this work, we addressed and analyzed monotonicity and error-boundness in differentiable sorting and ranking operators. Specifically, we focussed on differentiable sorting networks and presented a family of sigmoid functions that preserve monotonicity and bound approximation errors in differentiable sorting networks. This makes the sorting functions quasiconvex, and we empirically observe that the resulting method outperforms the state-of-the-art in differentiable sorting supervision.

ACKNOWLEDGMENTS & FUNDING DISCLOSURE

We warmly thank Robert Denk for helpful discussions. This work was supported by the Goethe Center for Scientific Computing (G-CSC) at Goethe University Frankfurt, the IBM-MIT Watson AI Lab, the DFG in the Cluster of Excellence EXC 2117 "Centre for the Advanced Study of Collective Behaviour" (Project-ID 390829875), and the Land Salzburg within the WISS 2025 project IDA-Lab (20102-F1901166-KZP and 20204-WISS/225/197-2019).

REPRODUCIBILITY STATEMENT

We made the source code and experiments of this work publicly available at github.com/Felix-Petersen/diffsort to foster future research in this direction. All data sets are publicly available. We specify all necessary hyperparameters for each experiment. We use the same model architectures as in previous works. We demonstrate how the choice of hyperparameter $\beta$ affects the performance in Figure 5. Each experiment can be reproduced on a single GPU.

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

## A  IMPLEMENTATION DETAILS

For training, we use the same network architecture as in previous works [1], [2], [4] and also use the Adam optimizer [20] at a learning rate of $3 \cdot 10^{-4}$. For Figure 5, we train for $100\,000$ steps. For Table 3, we train for $200\,000$ steps on MNIST and $1\,000\,000$ steps on SVHN. We preprocess SVHN as done by Goodfellow *et al.* [21].

### A.1  INVERSE TEMPERATURE $\beta$

For the inverse temperature $\beta$, we use the following values, which correspond to the optima in Figure 5 and were found via grid search:

| Method | $n$ | 3 | 5 | 7 | 9 | 15 | 32 | 16 | 32 |
|---|---|---|---|---|---|---|---|---|---|
| | odd-even / bitonic | oe | oe | oe | oe | oe | oe | bi | bi |
| $\sigma$ | Logistic | 79 | 30 | 33 | 54 | 32 | 128 | 43 | 8 |
| $\sigma \circ \varphi$ | Log. w/ ART | 15 | 20 | 13 | 34 | 16 | 29 | 28 | 26 |
| $f_{\mathcal{R}}$ | Reciprocal Sigmoid | 14 | 60 | 69 | 44 | 120 | 1140 | 124 | 76 |
| $f_{\mathcal{C}}$ | Cauchy CDF | $14.5\pi$ | $51\pi$ | $71\pi$ | $15\pi$ | $40\pi$ | $169\pi$ | $12\pi$ | $48.5\pi$ |
| | | 45.6 | 160.2 | 223.1 | 47.1 | 125.7 | 531.0 | 37.7 | 152.4 |
| $f_{\mathcal{O}}$ | Optimal Sigmoid | 6 | 20 | 29 | 32 | 25 | 124 | 17 | 25 |

## B  PROPERTIES OF $min$ AND $max$

The core element of differentiable sorting networks is the relaxation of the conditional swap operation, allowing for a soft transition between passing through and swapping, such that the sorting operator becomes differentiable. It is natural to try to achieve this by using soft versions of the minimum (denoted by $min$) and maximum operators (denoted by $max$). But before we consider concrete examples, let us collect some desirable properties that such relaxations should have. Naturally, $min$ and $max$ should satisfy many properties that their crisp / hard counterparts min and max satisfy, as well as a few others (for $a, b, c \in \mathbb{R}$):

**Symmetry / Commutativity.**  Since min and max are symmetric/commutative, so should be their soft counterparts: $min(a, b) = min(b, a)$ and $max(a, b) = max(b, a)$.

**Ordering.**  Certainly a (soft) maximum of two numbers should be at least as large as a (soft) minimum of the same two numbers: $min(a, b) \leq max(b, a)$.

**Continuity in Both Arguments.**  Both $min$ and $max$ should be continuous in both arguments.

**Idempotency.**  If the two arguments are equal in value, this value should be the result of $min$ and $max$, that is, $min(a, a) = max(a, a) = a$.

**Inversion.**  As for min and max, the two operators $min$ and $max$ should be connected in such a way that the result of one operator equals the negated result of the other operator applied to negated arguments: $min(a, b) = -max(-a, -b)$ and $max(a, b) = -min(-a, -b)$.

**Stability / Shift Invariance.**  Shifting both arguments by some value $c \in \mathbb{R}$ should shift each operator's result by the same value: $min(a + c, b + c) = min(a, b) + c$ and $max(a + c, b + c) = max(a, b) + c$. Stability implies that the values of $min$ and $max$ depend effectively only on the difference of their arguments. Specifically, choosing $c = -a$ yields $min(a, b) = min(0, b - a) + a$ and $max(a, b) = max(0, b - a) + a$, and $c = -b$ yields $min(a, b) = min(a - b, 0) + b$ and $max(a, b) = max(a - b, 0) + b$.

**Sum preservation.** The sum of $min$ and $max$ should equal the sum of $\min$ and $\max$: $min(a,b) + max(a,b) = \min(a,b) + \max(a,b) = a + b$. Note that sum preservation follows from stability, inversion and symmetry: $min(a,b) = min(a-b,0)+b = b-max(0,b-a) = b-(max(a,b)-a) = a + b - max(a,b)$

**Bounded by Hard Versions.** Soft operators should not yield values more extreme than their crisp / hard counterparts: $\min(a,b) \leq min(a,b)$ and $max(a,b) \leq \max(a,b)$. Note that together with ordering this property implies idempotency, vi7.: $a = \min(a,a) \leq min(a,a) \leq max(a,a) \leq \max(a,a) = a$. Otherwise, they cannot be defined via a convex combination of their inputs, making it impossible to define proper $argmin$ and $argmax$, and hence we could not compute differentiable permutation matrices.

**Monotonicity in Both Arguments.** For any $c > 0$, it should be $min(a + c, b) \geq min(a, b)$, $min(a, b + c) \geq min(a, b)$, $max(a + c, b) \geq max(a, b)$, and $max(a, b + c) \geq max(a, b)$. Note that the second expression for each operator follows from the first with the help of symmetry / commutativity.

**Bounded Error / Minimum Deviation from Hard Versions.** Soft versions of minimum and maximum should differ as little as possible from their crisp / hard counterparts. However, this condition needs to be made more precise to yield concrete properties (see below for details).

Note that $min$ and $max$ *cannot* satisfy associativity, as this would force them to be identical to their hard counterparts. Associativity means that $max(a, max(b, c)) = max(max(a, b), c)$ and $min(a, min(b, c)) = min(min(a, b), c)$. Now consider $a, b \in \mathbb{R}$ with $a < b$. Then with associativity and idempotency $max(a, max(a, b)) = max(max(a, a), b) = max(a, b)$ and hence $max(a, b) = b = \max(a, b)$ (by comparison of the second arguments). Analogously, one can show that if associativity held, we would have $min(a, b) = a = \min(a, b)$. That is, one cannot have both associativity and idempotency. Note that without idempotency, the soft operators would not be bounded by their hard versions. As idempotency is thus necessary, associativity has to be given up.

If $min$ and $max$ are to be bounded by the crisp / hard version and symmetry, ordering, inversion and stability (which imply sum preservation) hold, they must be convex combinations of the arguments $a$ and $b$ with weights that depend only on the difference of $a$ and $b$. That is,

$$
\begin{aligned}
min(a,b) &= f(b - a) \cdot a + (1 - f(b - a)) \cdot b \\
max(a,b) &= (1 - f(b - a)) \cdot a + f(b - a) \cdot b,
\end{aligned}
$$

where $f(x)$ yields a value in $[0, 1]$ (due to boundedness of $min$ and $max$ by their crisp / hard counterparts). Due to inversion, $f$ must satisfy $f(x) = 1 - f(-x)$ and hence $f(0) = \frac{1}{2}$. Monotonicity of $min$ and $max$ requires that $f$ is a monotonically increasing function. Continuity requires that $f$ is a continuous function. In summary, $f$ must be a continuous sigmoid function (in the older meaning of this term, i.e., an s-shaped function, of which the logistic function is only a special case) satisfying $f(x) = 1 - f(-x)$.

As mentioned, the condition that the soft versions of minimum and maximum should deviate as little as possible from the crisp / hard versions causes a slight problem: this deviation can always be made smaller by making the sigmoid function steeper (reaching the crisp / hard versions in the limit for infinite inverse temperature, when the sigmoid function turns into the Heaviside step function). Hence, in order to find the best shape of the sigmoid function, we have to limit its inverse temperature. Therefore, w.l.o.g., we require the sigmoid function to be Lipschitz-continuous with Lipschitz constant $\alpha = 1$.

## C  ADDITIONAL EXPERIMENTS

In Figure 6, we display additional results for more setting analogous to Figure 5.

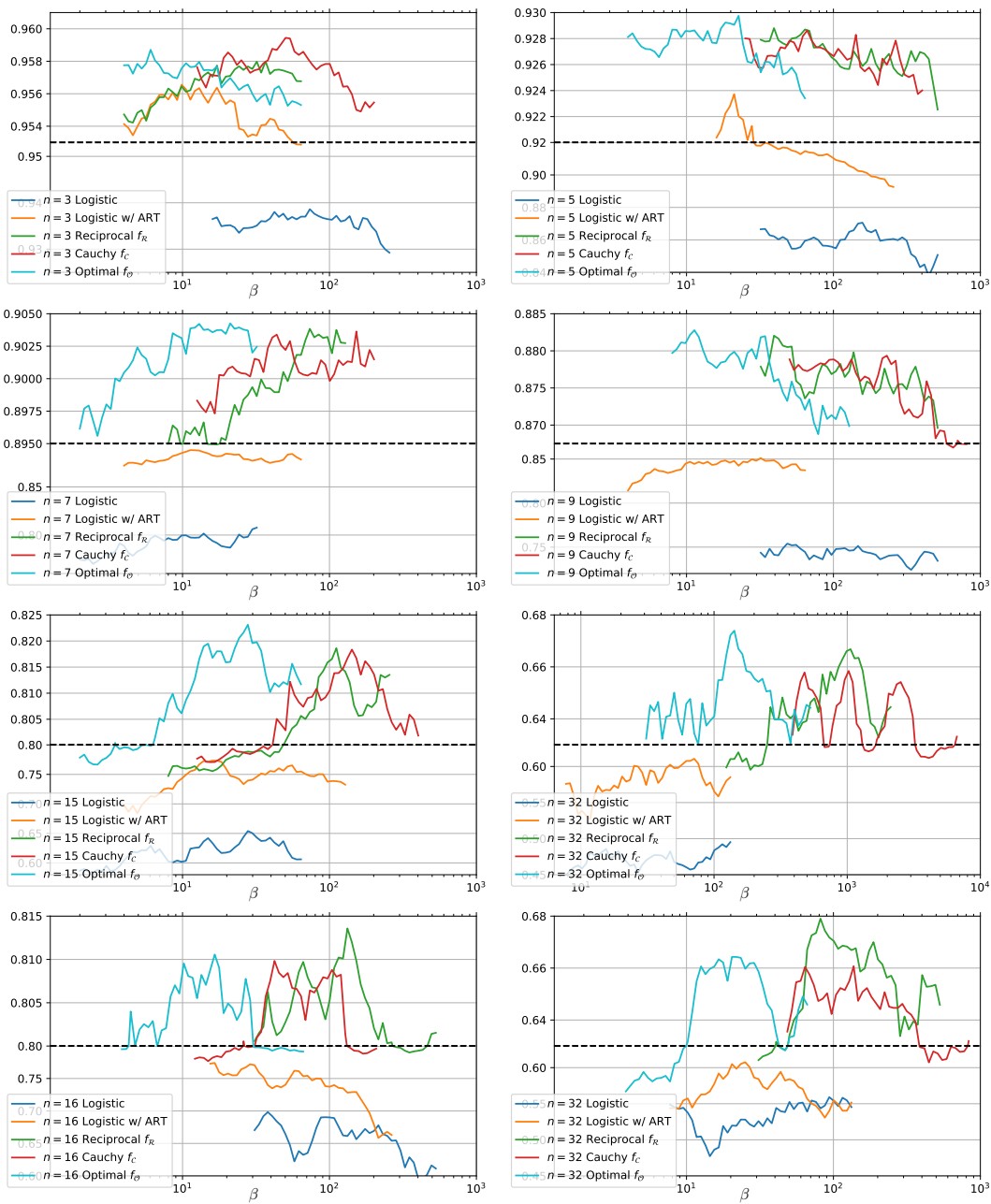

Figure 6: Additional results analogous to Figure 5. Evaluating different sigmoid functions on the sorting MNIST task for ranges of different inverse temperatures $\beta$. The metric is the proportion of individual element ranks correctly identified. In all settings, the monotonic sorting networks clearly outperform the non-monotonic ones. The first three rows use odd-even networks with $n \in \{3, 5, 7, 9, 15, 32\}$. The last row uses bitonic networks with $n \in \{16, 32\}$.

# D  ADDITIONAL PROOFS

**Theorem 9** (Error-Bounds of Diff. Sorting Networks). *If the error of individual conditional swaps of a sorting network is bounded by $\epsilon$ and the network has $\ell$ layers, the total error is bounded by $\epsilon \cdot \ell$.*

*Proof.* Induction over number $k$ of executed layers. Let $x^{(k)}$ be input $x$ differentially sorted for $k$ layers and $\mathbf{x}^{(k)}$ be input $x$ hard sorted for $k$ layers as an anchor. We require this anchor, as it is possible that $\mathbf{x}_i^{(k)} < \mathbf{x}_j^{(k)}$ but $x_i^{(k)} > x_j^{(k)}$ for some $i, j, k$.

Begin of induction: $k = 0$. Input vector $x$ equals the vector $x^{(0)}$ after 0 layers. Thus, the error is equal to $0 \cdot \epsilon$.

Step of induction: Given that after $k - 1$ layers the error is smaller than or equal to $(k-1)\epsilon$, we need to show that the error after $k$ layers is smaller than or equal to $k\epsilon$.

The layer consists of comparator pairs $i, j$. W.l.o.g. we assume $\mathbf{x}_i^{(k-1)} \leq \mathbf{x}_j^{(k-1)}$. W.l.o.g. we assume that wire $i$ will be the $\min$ and that wire $j$ will be the $\max$, therefore $\mathbf{x}_i^{(k)} \leq \mathbf{x}_j^{(k)}$. This implies $\mathbf{x}_i^{(k-1)} = \mathbf{x}_i^{(k)}$ and $\mathbf{x}_j^{(k-1)} = \mathbf{x}_j^{(k)}$. We distinguish two cases:

- $\left(\mathbf{x}_i^{(k-1)} \leq \mathbf{x}_j^{(k-1)} \text{ and } x_i^{(k-1)} \leq x_j^{(k-1)}\right)$    According to the assumption, $\left|x_i^{(k-1)} - x_i^{(k)}\right| \leq \epsilon$ and $\left|x_j^{(k-1)} - x_j^{(k)}\right| \leq \epsilon$. Thus, $\left|x_i^{(k)} - \mathbf{x}_i^{(k)}\right| \leq \left|x_i^{(k-1)} - \mathbf{x}_i^{(k-1)}\right| + \left|x_i^{(k-1)} - x_i^{(k)}\right| \leq (k-1)\epsilon + \epsilon = k\epsilon$.

- $\left(\mathbf{x}_i^{(k-1)} \leq \mathbf{x}_j^{(k-1)} \text{ but } x_i^{(k-1)} > x_j^{(k-1)}\right)$    This case can only occur if $\left|x_j^{(k-1)} - \mathbf{x}_i^{(k-1)}\right| \leq (k-1)\epsilon$ and $\left|x_i^{(k-1)} - \mathbf{x}_j^{(k-1)}\right| \leq (k-1)\epsilon$ because $\mathbf{x}_i^{(k-1)}$ and $\mathbf{x}_j^{(k-1)}$ have to be so close that within margin of error such a reversed order is possible. According to the assumption, $\left|x_j^{(k-1)} - x_i^{(k)}\right| \leq \epsilon$ and $\left|x_i^{(k-1)} - x_j^{(k)}\right| \leq \epsilon$. Thus, $\left|x_i^{(k)} - \mathbf{x}_i^{(k)}\right| \leq \left|x_j^{(k-1)} - \mathbf{x}_i^{(k-1)}\right| + \left|x_j^{(k-1)} - x_i^{(k)}\right| \leq (k-1)\epsilon + \epsilon = k\epsilon$. $\square$

**Theorem 11.** $\min_{f_\mathcal{R}}$ *and* $\max_{f_\mathcal{R}}$ *are monotonic functions with the sigmoid function* $f_\mathcal{R}$.

*Proof.* Wlog., we assume $a_i = x$ and $a_j = 0$.

$$min_{f_\mathcal{R}}(x, 0) = x \cdot f_\mathcal{R}(-x) = x \frac{1}{2}\left(\frac{x}{1 + |x|} + 1\right) \tag{14}$$

To show monotonicity, we consider its derivative / slope.

$$\frac{d}{dx} min_{f_\mathcal{R}}(x, 0) = \frac{d}{dx}\left(x\frac{1}{2}\left(\frac{x}{1 + |x|} + 1\right)\right) \tag{15}$$

$$= \frac{1}{2}\left(\frac{x}{1 + |x|} + 1\right) + x\frac{1}{2}\frac{d}{dx}\left(\frac{x}{1 + |x|} + 1\right) \tag{16}$$

$$= \frac{1}{2}\left(\frac{x}{1 + |x|} + 1\right) + x\frac{1}{2}\frac{d}{dx}\left(\frac{x}{1 + |x|}\right) \tag{17}$$

$$= \frac{1}{2}\left(\frac{x}{1 + |x|} + 1\right) + x\frac{1}{2}\frac{\frac{dx}{dx} \cdot (1 + |x|) - x \cdot \frac{d|x|+1}{dx}}{(1 + |x|)^2} \tag{18}$$

$$= \frac{1}{2}\left(\frac{x}{1 + |x|} + 1\right) + x\frac{1}{2}\frac{(1 + |x|) - x\,\mathrm{sgn}(x)}{(1 + |x|)^2} \tag{19}$$

$$= \frac{1}{2}\left(\frac{x}{1 + |x|} + 1\right) + x\frac{1}{2}\frac{1 + |x| - |x|}{(1 + |x|)^2} \tag{20}$$

$$= \frac{1}{2}\left(\frac{x}{1+|x|}+1\right) + x\frac{1}{2}\frac{1}{1+2|x|+|x|^2} \tag{21}$$

$$= \frac{1}{2}\left(\frac{x}{1+|x|}+1+\frac{x}{1+2|x|+|x|^2}\right) \tag{22}$$

$$= \frac{1}{2}\left(\frac{x(1+|x|)}{1+2|x|+|x|^2}+\frac{1+2|x|+|x|^2}{1+2|x|+|x|^2}+\frac{x}{1+2|x|+|x|^2}\right) \tag{23}$$

$$= \frac{1}{2}\left(\frac{2x+2|x|+x|x|+|x|^2+1}{1+2|x|+|x|^2}\right) \tag{24}$$

$$= \frac{1}{2}\left(\frac{2(x+|x|)+|x|(x+|x|)+1}{1+2|x|+|x|^2}\right) \tag{25}$$

$$\geq \frac{1}{2}\left(\frac{1}{1+2|x|+|x|^2}\right) \qquad \text{(because } x+|x|\geq 0) \tag{26}$$

$$> 0 \tag{27}$$

$\max_{f_{\mathcal{R}}}$ is analogous.

$\square$

**Theorem 12.** $\min_{f_{\mathcal{C}}}$ *and* $\max_{f_{\mathcal{C}}}$ *are monotonic functions with the sigmoid function* $f_{\mathcal{C}}$.

*Proof.* Wlog., we assume $a_i = x$ and $a_j = 0$.

$$min_{f_{\mathcal{C}}}(x,0) = x\cdot f_{\mathcal{C}}(-x) = x\cdot\left(\frac{1}{\pi}\arctan(-\beta x)+\frac{1}{2}\right) \tag{28}$$

To show monotonicity, we consider its derivative.

$$\frac{\partial}{\partial x}min_{f_{\mathcal{C}}}(0,x) = \frac{\partial}{\partial x}(f_{\mathcal{C}}(-x)\cdot x) = x\cdot\frac{\partial}{\partial x}f_{\mathcal{C}}(-x)+f_{\mathcal{C}}(-x)\cdot\frac{\partial}{\partial x}x$$

$$= x\cdot\frac{\partial}{\partial x}\left(\frac{1}{\pi}\arctan(-\beta x)+\frac{1}{2}\right)+\frac{1}{\pi}\arctan(-\beta x)+\frac{1}{2}$$

$$= x\cdot\frac{1}{\pi}\frac{-\beta}{1+(\beta x)^2}-\frac{1}{\pi}\arctan(\beta x)+\frac{1}{2}$$

$$= \frac{1}{2}-\frac{1}{\pi}\arctan(\beta x)-\frac{1}{\pi}\frac{\beta x}{1+(\beta x)^2}$$

$$= \frac{1}{2}-\frac{1}{\pi}\arctan(z)-\frac{1}{\pi}\frac{z}{1+z^2} \quad \text{(with } z = \beta x) \tag{29}$$

To reason about the derivative, we also consider the second derivative:

$$\lim_{z\to\infty}\frac{1}{2}-\frac{1}{\pi}\arctan(z)-\frac{1}{\pi}\frac{z}{1+z^2} = \frac{1}{2}-\lim_{z\to\infty}\frac{1}{\pi}\arctan(z)-\lim_{z\to\infty}\frac{1}{\pi}\frac{z}{1+z^2}$$

$$= \frac{1}{2}-\frac{1}{\pi}\frac{\pi}{2}-0=0 \tag{30}$$

$$\lim_{z\to-\infty}\frac{1}{2}-\frac{1}{\pi}\arctan(z)-\frac{1}{\pi}\frac{z}{1+z^2} = \frac{1}{2}-\lim_{z\to-\infty}\frac{1}{\pi}\arctan(z)-\lim_{z\to-\infty}\frac{1}{\pi}\frac{z}{1+z^2}$$

$$= \frac{1}{2}-\frac{1}{\pi}\frac{-\pi}{2}-0=\frac{1}{2}+\frac{1}{\pi}\frac{\pi}{2}-0=1 \tag{31}$$

For $z\in(-\infty,0]$: The derivative of $min_{f_{\mathcal{C}}}(0,x)$ converges to 1 for $z\to-\infty$ (Eq. 31).

For $z\in[0,\infty)$: The derivative of $min_{f_{\mathcal{C}}}(0,x)$ converges to 0 for $z\to\infty$ (Eq. 30).

$$\frac{\partial}{\partial z}\frac{1}{2}-\frac{1}{\pi}\arctan(z)-\frac{1}{\pi}\frac{z}{1+z^2} = -\frac{2}{\pi(1+z^2)^2} < 0 \tag{32}$$

The second derivative (Eq. 32) of $min_{f_{\mathcal{C}}}(0,x)$ is always negative.

Therefore, the derivative is always in $(0,1)$, and therefore always positive. Thus, $min_{f_{\mathcal{C}}}(0,x)$ is strictly monotonic. $\max_{f_{\mathcal{C}}}$ is analogous. $\square$

# E    ADDITIONAL DISCUSSION

*"How would the following baseline perform? Hard rank the predictions and compare it with the ground truth rank. Then, use their difference as the learning signal (i.e., instead of the gradient)."*

This kind of supervision does not converge, even for small learning rates and in simplified settings. Specifically, we observed in our experiments that the range of values produced by the CNN gets compressed heavily by training in this fashion. Also counteracting it by explicitly adding a term to spread it out again did not help, and training was very unstable. Despite testing various hyperparameters (learning rate, adaptation factor, both absolute and relative to the range of values in a batch or in the whole data set, spread factor, etc.) it did not work, even on toy data like single-digit MNIST with $n = 5$.

*"Could $\beta$ be jointly trained as a parameter with the model?"*

Yes, it could; however, in our experiments, we found that the entire training performs better if $\beta$ is fixed. If $\beta$ is also a parameter to be trained, its learning rate should be very small as it (i) should not change too fast and (ii) already accumulates many gradient signals as it is used many times.

