# OpenReview forum: "Monotonic Differentiable Sorting Networks"
_ICLR.cc/2022/Conference — ICLR 2022 Poster_

### Official Review · Reviewer_iDKW · 2021-10-28

**Correctness:** 4
**Technical Novelty And Significance:** 4
**Empirical Novelty And Significance:** 3
**Recommendation:** 8
**Confidence:** 3

**Main Review:**

Overall I think this is a good paper that should be accepted, as it presents a well motivated idea, contains strong theoretic results and supports their usefulness by improved experimental results on an existing problem setting (sorting images of numbers).

*Strengths*
- This paper presents a well motivated idea
- The paper is well structured and contains strong theoretical results
- Clear figures are very helpful for understanding the differences to other sigmoid functions
- Empirical results illustrate the improved performance gained from monotonicity
- Sensitivity to hyperparameter Beta and choice of sigmoid functional form is clearly illustrated, which shows monotonic networks clearly outperform non-monotonic alternatives.

*Weaknesses*
- The paper is a bit math heavy, which does hurt the flow of reading the paper a bit. I think some proofs could be deferred to the appendix, which would improve the overall readability of the paper.
- The experimental section is a bit limited as it only considers a sorting task. Related works consider additional experimental settings, for example quantile regression (Grover et al. 2019), which I think could also serve as a good test for this model.

*Minor comments/questions*
- Using italic vs roman font to distinguish relaxations can be confusing when skimming the paper. Maybe consider (additionally) using different names for the function. Maybe use the subscript min_sigma(a,b) in Eq. (1)?
- I am not sure, but do we need differentiability at all for the sorting task? I am curious how the following baseline would perform: simply sort the predictions and compare the rank with the ground truth rank and use their difference as learning signal ('gradient') for each item? It is obviously biased but it provides the correct direction, so in a sense is similar to monotonic differentiable sorting? (maybe this has already been tried or I'm missing something here)
- I'm a bit confused by 'for beta = 1' in Theorem 10, while beta appears in Eq. (13).
- In Figure 5, why are different ranges for Beta displayed for different functions? This would be good to clarify since (unlikely but in theory) the missing parts can change the picture. Also, given the explanation in the text, I suggest to align the horizontal axes so the figures can be compared.


**Summary Of The Paper:**

This paper proposes sigmoidal functions f for which, roughly speaking, x * f(-x) is non-decreasingly monotonic and bounded from above (this does not hold for the standard sigmoid). These are used to define continuous relaxations of min and max operations (not to confuse with softmax which is a relaxation for argmax) which are monotonic in their inputs.

By stacking multiple layers of these 'monotonic continuous conditional swaps' the authors obtain a sorting network for which all outputs are monotonic in all inputs. This ensures that the gradient is in the right direction when training a sorting network to sort data using only ground truth order.

Experiments on sorting 4-digit MNIST numbers or SVHN  show that the resulting network is able to predict the order more accurately than a number of alternatives, especially non-monotonic differentiable sorting networks.

**Summary Of The Review:**

This paper is a strong theoretic paper that proposes to replace the sigmoid function in differentiable sorting networks by one such that the network becomes monotonic and has bounded error, which guarantees the correct gradient direction when training sorting networks. Empirical results show the effectiveness on the task of sorting 4-digit MNIST numbers and SVHN.

While I think some additional experiments could be performed, I think this paper should be accepted given the well motivated idea, strong theoretical results, clear presentation illustrated by figures and the resulting empirical results.

---

> ### Author Response · Authors · 2021-11-19
> **Response to Reviewer iDKW**
>
> Thank you for your encouraging and constructive feedback.
> We tried to answer all of your questions below. Please don't hesitate to ask if anything is unclear or new questions come up.
>
> > "The paper is a bit math heavy, which does hurt the flow of reading the paper a bit. I think some proofs could be deferred to the appendix, which would improve the overall readability of the paper."
>
> We added transitional texts that put the proofs and definitions in context between them. For this, we deferred the proof of Theorem 9 into the supplementary material.
>
> > "The experimental section is a bit limited as it only considers a sorting task. Related works consider additional experimental settings, for example quantile regression (Grover et al. 2019), which I think could also serve as a good test for this model."
>
> Thank you for the hint. While we focus on the problem of sorting and ranking specifically, we believe that exploring other applications of differentiable sorting and ranking operators is a great idea for future work.
>
> > "Using italic vs roman font to distinguish relaxations can be confusing when skimming the paper. Maybe consider (additionally) using different names for the function. Maybe use the subscript min_sigma(a,b) in Eq. (1)?"
>
> Thanks for this suggestion, we changed it accordingly. We also made sure that at each instance of a relaxation a respective subscript (sigma, f, etc.) is added.
>
> > "I am not sure, but do we need differentiability at all for the sorting task? I am curious how the following baseline would perform: simply sort the predictions and compare the rank with the ground truth rank and use their difference as learning signal ('gradient') for each item? It is obviously biased but it provides the correct direction, so in a sense is similar to monotonic differentiable sorting? (maybe this has already been tried or I'm missing something here)"
>
> Thanks for bringing this up.
> We found in previous experiments that this kind of supervision did not converge, even for small learning rates and in simplified settings.
> Specifically, what we observed in our experiments was that the range of values produced by the CNN gets compressed heavily by training in this fashion.
> Even counteracting it by explicitly adding a term to spread it out again did not help, and training was very unstable.
> Despite playing around with various hyperparameters (learning rate, adaptation factor, both absolute and relative to the range of values in a batch or in the whole data set, spread factor etc.) it did not work at all, even on toy data like single-digit MNIST with n=5.
>
> > "I'm a bit confused by 'for beta = 1' in Theorem 10, while beta appears in Eq. (13)."
>
> In equation (13), we write the general form of the sigmoid function and with $\beta=1$ this function is 1-Lipschitz. We clarified this in the revision.
>
> > "In Figure 5, why are different ranges for Beta displayed for different functions? This would be good to clarify since (unlikely but in theory) the missing parts can change the picture. Also, given the explanation in the text, I suggest to align the horizontal axes so the figures can be compared."
>
> As the computation of many settings and with multiple seeds is costly, we started with a pre-evaluation run with a larger range than displayed to have a coarse assessment of the plot.
> We then selected the best performing range to cover them at a high resolution and multiple seeds.
> Also, not all sigmoid functions have the same derivative at position 0. That is, a Cauchy with $\beta=\pi$ more closely resembles the optimal function at $\beta=1$, which is why the red plot tends to be shifted to the right by half an order of magnitude.
>
> We will align the horizontal axes in the final version.

---

> > ### Comment · Reviewer_iDKW · 2021-11-26
> > **Thanks ; one more question about extra experiments**
> >
> > Thanks for the answers to my questions. It may be a good idea to clear some things up in the paper as well (regarding the differentiability and figure 5).
> >
> > While I still think this is a good paper, I do not agree that a different application of the same model should be future work. Can the monotonic differentiable sorting network be directly used for quantile regression or another task? If so, then (if code is available) I see no reason (beyond time constraints) not to test this, as it would really strengthen the paper, so I encourage the authors to do so in a next/final version. If not, then it would be good to clarify this since it limits the scope of the model.

---

> > > ### Author Response · Authors · 2021-11-29
> > > **Response**
> > >
> > > Thank you very much for your response!
> > >
> > > > It may be a good idea to clear some things up in the paper as well (regarding the differentiability and figure 5).
> > >
> > > We agree and will add a discussion on differentiability and will also clarify Figure 5 in the final version.
> > >
> > > > While I still think this is a good paper, I do not agree that a different application of the same model should be future work. Can the monotonic differentiable sorting network be directly used for quantile regression or another task? If so, then (if code is available) I see no reason (beyond time constraints) not to test this, as it would really strengthen the paper, so I encourage the authors to do so in a next/final version. If not, then it would be good to clarify this since it limits the scope of the model.
> > >
> > > You are right.
> > > We started reproducing the quantile (median) regression task from Grover et al. (2019).
> > > Here, we found that using a hard sorting operator performs better than any differentiable sorting method (NeuralSort, Sinkhorn Sort, our method).
> > > This also matches the results by Cuturi et al. (2019) who found that the test quantile error for the $\tau=$50% quantile (i.e., median) was better when using a hard sorting operator ($\epsilon=0$) on 5 out of 6 different data sets (cf. Table 2 in https://arxiv.org/pdf/1905.11885.pdf).
> > >
> > > Therefore, we also started reproducing the differentiable k-nearest neighbor task from Grover et al. (2019).
> > > As the discussion stage ends today (as per todays email) and the experiments are not finished, we propose to add the additional results to the final version.

---

> > > > ### Public Comment · ~Felix_Petersen1 · 2022-03-16
> > > > **follow up**
> > > >
> > > > Dear reviewer,
> > > >
> > > > As we were not able to reproduce the k-nearest neighbor (kNN) training (https://github.com/ermongroup/neuralsort), we instead extended the evaluation in Figure 5 by 4 additional plots / settings and added it to Figure 6 in the supplementary material.
> > > >
> > > > Best regards

---

### Official Review · Reviewer_vsFK · 2021-11-03

**Correctness:** 4
**Technical Novelty And Significance:** 3
**Empirical Novelty And Significance:** 3
**Recommendation:** 6
**Confidence:** 4

**Main Review:**

The paper presents an overview of differentiable sorting networks and the construction of soft min/max functions used in the swap operator. It follows that the construction of such operators need to follow a specific functional form, requiring the use of a generic sigmoid function. The paper presents a set of theorems to show that the derivative of this sigmoid function should be $\Omega(1/x^2)$ to guarantee end-to-end monotonicity, and should be $O(1/x^2)$ to guarantee end-to-end bounded error. A monotonic sorting network can be more easily trained (better behaved objectives) and has better logical consistency.

The paper introduces 3 candidates for the sigmoid functions that satisfy the conditions above. There is a limited set of experiments on multi-digit image ordering tasks constructed from MNIST and SVHN that show the new candidate sigmoids match or outperform all baselines across all network sizes. The paper also presents a study on the role of the inverse temperature hyper papermeter in the evaluation metric of the sorting network, suggesting that the optimal choice might be a function of the depth of the network rather than the number of elements being sorted.

Notes and comments:
- Can $\beta$ be a (shared) parameter to be trained jointly with the scoring model? That is assuming the objective is still well behaved for sgd.
- Typo in the definition of $h(x)$ in proof of Theorem 8.
- Could be helpful to reference Figure 2 in the intro to showcase the non-monotonic $min$ function constructed from logistic sigmoid.
- Some background on bitonic vs oe network would be helpful to have in the paper.


**Summary Of The Paper:**

This work introduces methods to construct differentiable sorting networks that are provably monotonic and have bounded error. To provide such guarantees, the paper derives a set of necessary and sufficient conditions on sigmoid function used for the soft swap operators in the network. It then studies a set of candidate sigmoid functions for such construction, detailing their monotonicity and error guarantees. The experimental results show that the use of monotonic networks can achieve SOTA results when applied to multi-digit ordering on images constructed from MNIST and SVHN.

**Summary Of The Review:**

Originality and significance: The main theoretical results presented in the paper (conditions on the monotonicity and bounded error) are novel to the best of my knowledge. The conclusions on using well suited sigmoid functions are interesting and relevant to the body of work on differentiable sorting networks.

Quality and clarity: The paper is well written and the presented results are easy to follow.

---

> ### Author Response · Authors · 2021-11-19
> **Response to Reviewer vsFK**
>
> Thank you for your encouraging and constructive feedback.
> We tried to answer all of your questions below. Please don't hesitate to ask if anything is unclear or new questions come up.
>
> > "Can $\beta$ be a (shared) parameter to be trained jointly with the scoring model? That is assuming the objective is still well behaved for sgd."
>
> Yes, it could; however, in our experiments we found that the entire training performs better if $\beta$ is fixed. If $\beta$ is also a parameter to be trained, its learning rate has to be very small as it (i) should not change too fast and (ii) already accumulates many gradient signals as it is used many times.
>
> Thanks for pointing out the typo.
>
> > "Could be helpful to reference Figure 2 in the intro to showcase the non-monotonic $\mathit{min}$ function constructed from logistic sigmoid."
>
> Thanks for the suggestion, we included it.
>
> > "Some background on bitonic vs oe network would be helpful to have in the paper."
>
> We extended the discussion on sorting networks in Section 3.
> However, as we propose a general method for any sorting networks architecture, we would like to refer to primary sources for additional information on the specific architectures ([15, 16] in the revised paper).

---

> > ### Comment · Reviewer_vsFK · 2021-11-28
> > **Comment**
> >
> > Thanks for answering my questions and concerns. It could be helpful to briefly mention your experience with trainable $\beta$ in the paper or the appendix.

---

> > > ### Author Response · Authors · 2021-11-29
> > > **Response to Comment**
> > >
> > > Thank you very much for your response!
> > > We agree and we will add a discussion on trainable $\beta$ in the final version.

---

### Official Review · Reviewer_QCdx · 2021-11-04

**Correctness:** 4
**Technical Novelty And Significance:** 2
**Empirical Novelty And Significance:** 2
**Recommendation:** 6
**Confidence:** 3

**Main Review:**

This is a nice contribution, the paper is well-written, the interest in finding mononicity is well-explained and motivated; it is no surprise then that we get better results.

I do believe on its current form the paper is a bit incremental and lacking substance which motivate my judgement. I do believe all of the points are addressable.

1)On the one hand, the theoretical findings, although they are meaningful, as mostly straightforward observations. Framing them as theorems appears to me as an overstatement. Proofs are simple and I would recommend them moving them to the appendix.

2)It takes 7 pages to get to the experiment section. This is consistent with my belief that experimental validation is weak. I understand it makes sense to have thorough discussion about functions, moninicity, etc, as authors excel at, but I believe that experiments should be extended. Other papers, such as https://arxiv.org/pdf/2105.04019.pdf provide a much more thorough experimental validation. Why this is not the case here? What happens with large n?

3)I appreciate environmental concerns by authors. However, this raises the point about scalability. Authors should comment on how scalable the proposed method is, hopefully making claims about computational complexity and/or reporting times. Is the proposed method as scalable as alternatives?

4)General interest. Sorting networks are a nice construction but I wonder whether the impact of this area of research is doomed to remain quite limited, or if it could produce impact in broader areas of machine learning as well. It would be good if authors can elaborate on broader impact of their work. As currently stated, beyond the genuine intellectual interest that these constructions may spark, this work appears a bit focused on a niche and with a compromised capacity of reaching a broader audience (I believe this is shared by all other cited works on sorting networks, though).

**Summary Of The Paper:**

This paper introduces monotic differentiable sinkhorn networks. Authors argue that mononicity is a desired property because it leads to better bounds in errors. Authors show their proposed approach beats the state of the art on sorting MNIST and SVHN digits

**Summary Of The Review:**

Nice contribution. Needs more empirics

---

> ### Author Response · Authors · 2021-11-19
> **Response to Reviewer QCdx**
>
> Thank you for your encouraging and constructive feedback.
> We tried to answer all of your questions. Please don't hesitate to ask if anything is unclear or new questions come up.
>
> > "1)On the one hand, the theoretical findings, although they are meaningful, as mostly straightforward observations. Framing them as theorems appears to me as an overstatement. Proofs are simple and I would recommend them moving them to the appendix."
>
> Thank you for appreciating the simplicity of our proofs.
> Using definitions and theorems allows us to structure the theoretical insights well and helps the reader to easily see the core theoretical results we obtained.
> The proofs do not only demonstrate that the theorems hold, but also provide insights into the underlying reasoning of our approach.
>
> We decided to move the proof of Theorem 9 to the appendix and added some additional information between the definitions and theorems to give them more context.
>
> > "2)It takes 7 pages to get to the experiment section. This is consistent with my belief that experimental validation is weak. I understand it makes sense to have thorough discussion about functions, moninicity, etc, as authors excel at, but I believe that experiments should be extended. Other papers, such as https://arxiv.org/pdf/2105.04019.pdf provide a much more thorough experimental validation. Why this is not the case here? What happens with large n?"
>
> Thank you for noticing that our paper is more theoretical than [A].
> Our goal was indeed to focus on the theoretical properties **and** demonstrate practical characteristics of monotonic differentiable sorting networks.
> We would like to point out that Table 3 and especially Figure 5 provide an in-depth evaluation of the proposed methods.
> Figure 5 demonstrates the performance of each method for a much larger range of hyperparameters and different methods than any evaluation by [A], (who select heuristic-based individual inverse temperatures).
> We agree that more experiments are always helpful.
> Due to the limited time of the response period and extended training time for larger $n$, we propose to publish code as well as results for larger $n$ on the project website.
>
> > "3)I appreciate environmental concerns by authors. However, this raises the point about scalability. Authors should comment on how scalable the proposed method is, hopefully making claims about computational complexity and/or reporting times. Is the proposed method as scalable as alternatives?"
>
> The proposed method has the same scalability as differentiable sorting networks with the logistic sigmoid function [A].
> This is because all of our employed sigmoid functions can easily be computed in closed form, therefore no computational overhead in comparison to the logistic sigmoid functions is measurable.
> We added a complexity discussion in the revision in Section 6.
>
> > "4)General interest. Sorting networks are a nice construction but I wonder whether the impact of this area of research is doomed to remain quite limited, or if it could produce impact in broader areas of machine learning as well. It would be good if authors can elaborate on broader impact of their work. As currently stated, beyond the genuine intellectual interest that these constructions may spark, this work appears a bit focused on a niche and with a compromised capacity of reaching a broader audience (I believe this is shared by all other cited works on sorting networks, though)."
>
> We understand that the impact of differentiable sorting networks has thus far been quite limited; however, as they provide a very competitive method for differentiable sorting and ranking, we believe them to have a large potential.
>
> In the domain of recommender systems, [D] propose differentiable ranking metrics, and [E] propose PiRank, a learning-to-rank method using differentiable sorting.
> Furthermore, there are works exploring differentiable sorting-based top-k for applications such as differentiable image patch selection [B], differentiable k-nearest-neighbor [F, G], top-k attention for machine translation [F], and differentiable beam search methods ([C, F]).
> We also believe that differentiable sorting can be useful in the context of self-supervised learning as elements such as contrastive loss are technically based on learning-to-rank.
>
> Thank you for asking us to collect related works to better shape the description of the broader impact of our work. We included the additional references in the revision.
>
> ---
>
> **[A]** https://arxiv.org/pdf/2105.04019.pdf
>
> **[B]** https://arxiv.org/abs/2104.03059
>
> **[C]** https://arxiv.org/abs/1708.00111
>
> **[D]** https://arxiv.org/abs/2008.13141
>
> **[E]** https://arxiv.org/abs/2012.06731
>
> **[F]** https://arxiv.org/abs/2002.06504
>
> **[G]** https://arxiv.org/pdf/1903.08850.pdf

---

> > ### Comment · Reviewer_QCdx · 2021-11-29
> > **Thank you for your response**
> >
> > Thanks to the authors for their thorough response!

---

### Official Review · Reviewer_28Ue · 2021-11-04

**Correctness:** 3
**Technical Novelty And Significance:** 3
**Empirical Novelty And Significance:** 2
**Recommendation:** 5
**Confidence:** 2

**Main Review:**

First of all, I want to say that I am not an expert on differentiable sorting and so my opinion reflects the one from a novice but curious reader.
## Pros
- The idea of enforcing monotonicity is sound.
- Empirical results demonstrate a real gain from using monotonic sorting networks.

## Cons
- Overall, I have the feeling this paper was written in a rush. Section 4 is not very readable as it is just a series of theorems and proofs without any flow between them. Proofs are also very poorly shaped and sometimes does not seem very rigorous (e.g. proof of theorem 5 is missing some strict logical relationships between some steps). Moreover, as a novice it quite difficult to understand how Figure 5 should be analysed, and the long caption does not help much.
- You provide some necessary conditions for monotonicity but not any sufficient one. And you do not prove explicitly that the functions you introduced are monotonic.
- It is not very clear to me why you introduce Fc and Fr.
- As fast sort is monotonic as well why doesn't it work better?
- What are the missing entries in table 3?
- Quasi convexity is mentioned once in the intro and second time in the conclusion but never elsewhere. This contributed to my feeling that the paper was not well polished. Maybe this is something that could be stated more explicitly and motivated somewhere in the section 4 or 5?
## Additional remarks:
Shouldn't P in Figure 1 be doubly stochastic? (second columns does not sum up to 1).

**Summary Of The Paper:**

This paper presents monotonic differentiable sorting networks. It argues that monotonicity is lacking in most differentiable sorting networks whereas this may lead to incorrect gradient signs and inconsistent outputs. The first part of the paper motivates the problem and provides a brief review of other differentiable sorting networks. Then the paper provides a necessary condition for monotonicity and study bounds on the error of continuous conditional swaps. Finally, experiments demonstrate that monotonic swaps outperform non-monotonic ones for training a network to order MNIST, especially on larger numbers.

**Summary Of The Review:**

I believe the idea proposed in this paper is interesting. However, the paper presentation should be improved and further discussion is required to make it a strong paper.

---

> ### Author Response · Authors · 2021-11-19
> **Response to Reviewer 28Ue**
>
> Thank you very much for your review.
> We tried to answer all of your questions. Please don't hesitate to ask if anything is unclear or new questions come up.
>
> > "Section 4 is not very readable as it is just a series of theorems and proofs without any flow between them."
>
> We added transitional texts that put the proofs and definitions in context between them. For this, we deferred the proof of Theorem 9 into the supplementary material.
>
> > "Proofs are also very poorly shaped and sometimes does not seem very rigorous (e.g. proof of theorem 5 is missing some strict logical relationships between some steps)."
>
> Could you please clarify what exactly is not rigorous, and what is the strict logical relationship that is missing in theorem 5?
>
> > "Moreover, as a novice it quite difficult to understand how Figure 5 should be analysed, and the long caption does not help much."
>
> As the temperature $\beta$ is an important parameter for differentiable sorting networks, it is important to evaluate the method for different choices of this parameter in order to paint the full picture of the performance.
> This is what we plot on the x-axis.
> On the y-axis, we plot an evaluation metric where larger values are better.
> The plots display both the performance for different hyperparameter choices but also the sensitivity to the hyperparameter.
> Important aspects in the plot are:
>
> * The peak performance, i.e., the largest point of each line, which should be as large as possible.
> * Whether the peak is pointed or rather flat, here a rather flat peak makes it easier to find a good hyperparameter.
> * If a line is above another line, this means that the method is better regardless of the hyperparameter choice.
>
> > "You provide some necessary conditions for monotonicity but not any sufficient one."
>
> The sufficient condition for monotonicity is $\textit{min}'_f(x,0) \geq 0$ for all $x$. We included this explicitly in the revision.
>
> > "And you do not prove explicitly that the functions you introduced are monotonic."
>
> We apologize for this oversight. We did prove monotonicity for the reciprocal and the Cauchy sigmoid functions but did not end up including it in the draft.
> We now included the proofs in the revision. We would like to point out that we did prove the monotonicity of the optimal sigmoid function via its proof-by-construction.
>
> > "It is not very clear to me why you introduce Fc and Fr."
>
> We introduce $f_\mathcal{R}$ as it is a natural and simple choice for a monotonic and bounded function.
> We introduce $f_\mathcal{C}$ as it has an interesting probabilistic interpretation and is a smooth function with a quite small bound, and has a simple closed form solution. For this kind of simplicity and smoothness, it is hard to find a function with a much smaller error-bound (there are some, but they don't reduce the bound by much and are quite involved functions, some of which do not even have a closed form solution).
>
> > "As fast sort is monotonic as well why doesn't it work better?"
>
> While FastSort is indeed monotonic, it does not provide a bounded error.
> Without a bounded error, the error of FastSort diverges to $\infty$ (see Figure 4).
> In addition, the FastSort method (as it does not provide a differentiable permutation matrix) cannot make use of the cross entropy loss and instead has to use the somewhat less effective MSE sorting loss.
>
> > "What are the missing entries in table 3?"
>
> * For the non-sorting network methods, there is no "bitonic" because they do not have a sorting network choice.
> * For FastSort w/ 32, the training did not work.
> * For SVHN, we omitted $n=32$ as the evaluation would be fairly computationally costly because the SVHN CNN is rather large and propagating the images through the CNN is expensive (the effective cost is thus linear in $n$, therefore omitting 32 cuts the total evaluation cost by more than half).
>
> > "Quasi convexity is mentioned once in the intro and second time in the conclusion but never elsewhere. This contributed to my feeling that the paper was not well polished. Maybe this is something that could be stated more explicitly and motivated somewhere in the section 4 or 5?"
>
> Thank you for the suggestion, we included it in the discussion in Section 4.1.
>
> > "Shouldn't P in Figure 1 be doubly stochastic? (second columns does not sum up to 1)."
>
> It is doubly stochastic, however, due to rounding (for display) an error accumulated, which is why the rounded variant does not sum up to exactly one.
> In the revision, we modified the rounding to avoid this confusion.

---

### Author Response · Authors · 2021-11-19
**Revision**

Dear reviewers and AC,

Thank you all for your time investment and for your comments and constructive criticisms.
The main changes of the revision are:

* Based on feedback by *28Ue*, *QCdx*, and *iDKW*, we added clarifications between the definitions, theorems, and corollary of Section 4.1 and deferred the proof of Theorem 9 to the supplementary material.
* Based on feedback by *28Ue*, we included proofs for the montonicity of $f_\mathcal{R}$ and $f_\mathcal{C}$ in the supplementary material. Also, we extended the discussion in Section 4.1 by quasiconvexity.
* Based on feedback by *QCdx*, we included a discussion on "Applications and Broader Impact" in the related work section, and also include a discussion on the computational complexity in Section 6.
* Based on feedback by *vsFK*, we extended the discussion of sorting network architectures and added a reference to Figures 2 and 4 in the introduction.

We have marked all new sentences and paragraphs in red to make it easier to find the changes.
(Typos, notation, as well as the proofs in the SM have not been colored.)
More details and information of other changes including improved notation and typos can be found in the respective author responses for each reviewer.

Please don't hesitate to ask if anything is unclear or new questions come up.

Best regards,

The Authors

---

### Public Comment · ~Felix_Petersen1 · 2022-04-23
**Code / Project Page**

The code and project page is available under: https://github.com/Felix-Petersen/diffsort

---

### Decision · Program_Chairs · 2022-01-20

**Decision:**

Accept (Poster)

**Comment:**

This submission presents an interesting contribution on differentiable sorting, providing an analysis of monotonicity for these operations.

The reviewers overall argue for acceptance.